# Acoustic Emission and Artificial Intelligence Procedure for Crack Source Localization

**DOI:** 10.3390/s23020693

**Published:** 2023-01-07

**Authors:** Jonathan Melchiorre, Amedeo Manuello Bertetto, Marco Martino Rosso, Giuseppe Carlo Marano

**Affiliations:** 1Department of Structural, Geotechnical and Building Engineering (DISEG), Politecnico di Torino, Corso Duca Degli Abruzzi, 24, 10128 Turin, Italy; 2College of Civil Engineering, Fuzhou University, Fuzhou 350108, China

**Keywords:** acoustic emission, artificial neural network, Akaike Information Criterion (AIC), source location, seismic signals, crack location, sound event detection

## Abstract

The acoustic emission (AE) technique is one of the most widely used in the field of structural monitoring. Its popularity mainly stems from the fact that it belongs to the category of non-destructive techniques (NDT) and allows the passive monitoring of structures. The technique employs piezoelectric sensors to measure the elastic ultrasonic wave that propagates in the material as a result of the crack formation’s abrupt release of energy. The recorded signal can be investigated to obtain information about the source crack, its position, and its typology (Mode I, Mode II). Over the years, many techniques have been developed for the localization, characterization, and quantification of damage from the study of acoustic emission. The onset time of the signal is an essential information item to be derived from waveform analysis. This information combined with the use of the triangulation technique allows for the identification of the crack location. In the literature, it is possible to find many methods to identify, with increasing accuracy, the onset time of the P-wave. Indeed, the precision of the onset time detection affects the accuracy of identifying the location of the crack. In this paper, two techniques for the definition of the onset time of acoustic emission signals are presented. The first method is based on the Akaike Information Criterion (AIC) while the second one relies on the use of artificial intelligence (AI). A recurrent convolutional neural network (R-CNN) designed for sound event detection (SED) is trained on three different datasets composed of seismic signals and acoustic emission signals to be tested on a real-world acoustic emission dataset. The new method allows taking advantage of the similarities between acoustic emissions, seismic signals, and sound signals, enhancing the accuracy in determining the onset time.

## 1. Introduction

The need to preserve and maintain historical buildings, structural heritage and civil infrastructures combined with improved safety standards has led to the increasing use of structural health monitoring (SHM) techniques [1,2,3,4,5,6,7,8]. The adoption of reliable and rigorous monitoring systems is fundamental to reducing maintenance costs and at the same time extending the service life of the existing structures. In particular, to properly define the health state of existing structures, many techniques for identifying and locating damage have been developed in literature [9,10]. The acoustic emission (AE) techniques are widely adopted among the monitoring methods because they are non-destructive techniques that allow the passive monitoring of structures [11,12]. The use of these techniques involves the installation of piezoelectric sensors able to record the transient ultrasonic wave that propagates in the material and that is triggered by the sudden release of elastic energy at the moment of crack formation. The sensors can convert the elastic energy of the ultrasonic wave into an electric waveform that can be digitized and analyzed [13]. By studying and investigating the recorded signals, it is possible to obtain indirect information about the nature of the cracking pattern formation and evolution. Over the last years, many techniques have been developed for the localization, characterization, and quantification of damage from the study of acoustic emission signals [13,14,15,16]. One of the main applications of structural monitoring through the acoustic emission technique is the localization of the crack. The timely identification of the fracturing source allows a prior understanding of the possible damage mechanism [17]. In this way, it is possible to take action promptly with targeted maintenance work that will increase safety and extend the service life of the building. To localize the damage, the accuracy in the determination of the onset time is a crucial aspect since it affects the precision of the crack event location. The onset time of an acoustic emission signal can be defined as the moment in which the elastic wave reaches the sensor positions for the first time [18]. In general, the onset time is referred to as the point at which the difference between the signal and the background noise occurs for the first time [18]. The monitoring process is performed by a continuous recording system. The identification of the onset time must be performed automatically because of the large amount of recorded data to be analyzed. For this reason, several techniques for automatic detection of onset time have been presented over the years. Most of these methods have been developed in the field of seismology. The application of these techniques to acoustic emission analysis is possible because of their relationship to seismological studies [19]. Indeed, it can be asserted that AE and earthquakes are the outcomes of the same phenomenon released at different scales. One of the most basic methods for onset time detection is to use the amplitude static or dynamic threshold. In this method, the onset time is defined as the moment in which the amplitude of the analyzed signal exceeds the set threshold value [20]. Even if this method has been successfully applied in seismic analysis [21], in the case of acoustic emission monitoring, it can be difficult to set appropriate amplitude limit values maintaining a good accuracy level. In this field of application, the amplitudes of the analyzed waveforms may have a similar order of magnitude as those characterizing the background noise. This results in a very rough approximation of the onset time that can lead to a mislocalization of the crack source. The threshold approach can be improved by considering the use of complex dynamic threshold values. The main idea is to update the threshold on the basis of the average acoustic noise amplitude. This approach is the basis of the STA/LTA technique (STA—short-term average, LTA—long-term average) [22]. In this method, the limiting value is defined based on the comparison between STA, which is a measure of the instantaneous amplitude of the signal, and LTA, which contains information about the average amplitude of the background noise. Although the STA/LTA technique has proven promising in the analysis of seismic signals, it does not produce sufficiently accurate results in the case of acoustic emissions due to the fact that signal and noise are often to be found in the same amplitude range. Other approaches have been proposed to determine the dynamic threshold by studying the spectrograms of the signals [23]. Additionally, in this case, the methods have proved to be more effective in the case of seismic signals than for an application in the acoustic emission field. This is because acoustics and background noise are often found in the same frequency range, especially in case that they are applied to structural materials such as concrete and steel. Alternative methods have been presented for identifying onset time in signals characterized by low signal-to-noise (S/N) ratios. In particular, Boschetti et al. [24] proposed a method based on the variation in fractal dimension along the trace. The proposed fractal-based algorithm proved to be accurate, even in the presence of significant noise. Although this method can tolerate noise up to 80% of the average signal amplitude, it results in being considerably slower than other methods and is not suitable for real-time applications. In order to overcome the flaws that characterize the above-presented methodologies, another approach for onset time determination based on studying the signals as autoregressive (AR) models, was proposed. These models expect that the output variable is linearly dependent on the previously assumed values and on a stochastic term. The application of the Akaike Information Criterion (AIC) [25] to autoregressive models allows to properly divide the acoustic emission signals into two stationary datasets before and after the onset time. In general, by minimizing the AIC for a fixed order AR process, it is possible to obtain the point that determines the separation of the two time series. In this way, the most probable onset time is determined. The drawback of the presented methods is that is not possible to check the validity of the detected times. Hence, it is required that the adopted method should be sufficiently accurate to avoid the possibility of false recognitions of arrival times. Otherwise, it must be defined a postprocessing method that allows to automatically check the validity of the detected results. Carpinteri et al. [26] proposed an improved AIC procedure based on the estimation of the degree of accuracy of AE signals by the second derivative of the AIC function and by a parameter related to the propagation velocity of the elastic waves. In recent decades, the rise of machine learning and, more generally, artificial intelligence techniques, have led to the development of new perspectives in all engineering fields. The new methods, based on data-driven approaches, have shown excellent results, as they allow for taking into account complex aspects of the problems studied. These new algorithms have been applied for damage identification and localization purposes by many authors [27,28]. In particular, artificial neural networks were used for the automatic identification of onset time and subsequent damage localization [29,30]. In this paper, a new data-driven approach for the identification of the onset time is presented. The new method involves using a convolutional recurrent (CRNN) neural network for the onset time identification. The use of CNN for crack identification in structural materials has already been explored in the literature [31,32,33]. In this work, a neural network designed for sound event detection (SED) was trained using three different datasets consisting of seismic signals and/or acoustic emission signals. The selected neural architecture proved to be more effective than other CNN models in detecting the onset time of acoustic emission signals [34]. The dataset of seismic accelerograms was procured from the Italian Accelerometric Archives (ITACA) database. It contains 410 seismic events time series that were analyzed to manually define the onset time. The network receives inputs in terms of spectrograms to account for intrinsic features in both the time and frequency domains. After the training, the CRNN was used to identify the source location of a pencil lead break (PLB) test on the face of a concrete block. The PLB, also known as the Hsu–Nielsen source, is an artificial method of generating acoustic emission (AE) signals, which can roughly represent a source of acoustic emission damage. The data used by the authors to test the method can be found in reference [35]. In this research, the main idea is to exploit the relationships among sound waves, seismic waves, and acoustic emissions. In this scenario, it was demonstrated how using an architecture aimed at SED with the addition of seismic signal data, it was possible to enhance the accuracy in determining onset time.

## 2. Crack Localization: Triangulation Procedure

Several models for the source localization using acoustic emission signals have been proposed in the literature [36,37,38]. In this paper, the proposed method for crack localization is based on the assumption that the elastic wave travels directly from the crack source to the acoustic sensors. Thus, it implies that the wave path is represented by the straight line connecting the point where the crack occurs and the sensors. The shortest wave path model is a simplifying assumption commonly used in the field of acoustic emissions [26,39]. This hypothesis can be applied especially considering that the proposed method is intended to be used to detect the onset time of P-waves (longitudinal waves). In general, while the onset times of the P-waves (longitudinal waves) and S-waves (shear waves) can be used for crack characterization, only first wave onset times (P-wave times) are usually employed since they are less affected by multiple side reflections, structural noise and sensor response that will interfere with the later phases [26]. More complex models that take into account possible geometrical and material irregularities can be found in the literature [40]. In this preliminary study, the simplest model was chosen to focus on the presented method for the identification of the onset time. Under this assumption, the distance between the crack source location *S*, defined by the coordinates (x,y,z), and the known position of the i-th sensor $S_{i}$, defined by the coordinates $\left(x_{i},y_{i},z_{i}\right)$, can be calculated as
(1)$$|S-S_{i}|=\sqrt{{\left(x-x_{i}\right)}^{2}+{\left(y-y_{i}\right)}^{2}+{\left(z-z_{i}\right)}^{2}}$$

Considering the medium in which the elastic wave is transmitted to be homogeneous, such a distance can be also defined by the kinematic relationships. Defining $v_{p}$ as the speed of the wave in the medium, $t_{0}$ as the instant of crack occurrence, and $t_{i}$ as the onset time for the transducer *i*, it is possible to define
(2)$$|S-S_{i}|=v_{p}\left(t_{i}-t_{0}\right)$$

Thus, combining the two Equations (Equation 1) and (Equation 2), a new equation is obtained:(3)$$\sqrt{{\left(x-x_{i}\right)}^{2}+{\left(y-y_{i}\right)}^{2}+{\left(z-z_{i}\right)}^{2}}=v_{p}\left(t_{i}-t_{0}\right)$$

The position of the sensors $\left(x_{i},y_{i},z_{i}\right)$ is generally known. The onset time $t_{i}$ can be obtained by the registered signal and thus the equation is characterized by five unknowns $\left[x,y,z,v_{p},t_{0}\right]$. If a second transducer *j* is considered, it is possible to apply Equation (Equation 3) and to eliminate the variable $t_{0}$ by applying the subtraction method between the equations defined for *i* and *j*:(4)$$|S-S_{i}\left|-\right|S-S_{j}|=v_{p}\Delta t_{ij}$$

Therefore, the remaining unknowns of the problem are the position of the crack and the transmission velocity of the elastic wave in the studied medium. The position of *S* is a problem that can be solved exactly if there are enough equations of the type (Equation 4). A minimum of five acoustic sensors are required to calculate the exact solution for the problem and univocally identify the source crack position. Finally, it is possible to solve the nonlinear equation system by applying an iterative algorithm. In general, it is preferable to use a larger number of transducers in order to apply the least squares approach to minimize and calculate the error in the obtained results [18,41].

## 3. Onset Time Detection

The onset time detection is a key step in the crack localization process. Although it is easy for an experienced technician to detect the time of arrival in an acoustic signal, it is essential to develop techniques that are capable of automatically accomplishing this task. Indeed, monitoring through the acoustic emission technique allows the recording of a large amount of data, and manual processing results is an overly time-consuming job. The precision in crack localization is strongly influenced by the accuracy in the determination of the onset time. In addition, in some applications, the automatic processing of the recorded signals should be carried out in quasi-real-time manner. In these cases, an “a posteriori” check in order to identify appreciable errors in the measurements is not possible. For these reasons, onset time identification techniques need to be simultaneously automatic, accurate, fast, and reliable. In this paper, two methods for automatic onset time identification are presented and compared in terms of accuracy.

### 3.1. Improved AIC Picker

The improved AIC picker method, presented by Carpinteri et al. [26] and represented in Figure 1, is based on the application of the Akaike Information Criterion (AIC) to autoregressive models. The method involves using the AIC to find the best point to divide the analyzed signal into two time series. The division must be made at the onset time. The time series before the onset time is the one in which only the background noise is present, while the other after the onset point also includes the acoustic emission signal. The presented technique seems to be much more effective when applied to a selected time series in which only one acoustic signal was recorded [42]. The selection of the time window is done by a threshold amplitude method, which allows for obtaining an initial estimate of the onset time. Since $x_{t}$ is the time series, the threshold amplitude is defined by comparing the average amplitude of a translated set of ten data points with the average amplitude multiplied by four of the signal from the first data point to a *k* point, as in the Equation (Equation 5).
(5)$$\frac{\left(\sum_{t=k+1}^{10}\left|x_{t}\right|\right)}{10}\geq 4\frac{\left(\sum_{t=1}^{k}\left|x_{t}\right|\right)}{k}$$

The acoustic signal is analyzed, iteratively incrementing the value of *k*. The procedure is reiterated until obtaining the first value of *k* for which the Equation (Equation 5) is satisfied, $k_{0}$. The calculated value, $k_{0}$, represents the first estimate of onset time. In general, point $k_{0}$ is always located after the actual onset time. For this reason, the time window in interval $\left[1,k_{0}\right]$ contains the actual onset time, and the improved AIC picker method can be applied to it. Application of the method allows the definition of a new value $k_{1}$ that represents a more precise estimation of the onset time. The ultimate time window is defined as centered on the value $k_{1}$ and a time interval width equal to 2Δk sampled points. This results in the time window in which to definitively apply the improved AIC selection method. In general, the value of Δk is defined as a function of the sampling rate. In [26], Δk=3000 samples in the case of a test performed with a sampling frequency of 10 MHz is recommended. Finally, the onset time is calculated by applying the improved AIC picker method to the established time window. In total, the sequential application of a threshold-based method to find the first attempt onset time $k_{0}$ and the double application of the improved AIC picker method allow for redundancy and a consequent increase in the accuracy of the obtained measurement. The onset time is calculated as the time instant corresponding to the minimum point of the Akaike Information Criterion [43]. Therefore, the definition of an appropriate time window over which to apply the method is critical. In fact, in order to correctly identify the onset time of all signals, a well-defined minimum must be associated with each of them. The AIC Equation (Equation 6), derived from [25], can be defined as a function of the number of signal parameters *k* and the maximum value of the likelihood function for the estimated model:(6)AIC=−2ln(L)+2k

The proposed method assumes that the analyzed time window is composed of two different stationary time series: the first with only the environmental noise and the second with the acoustic signal. The technique consists of splitting the time series $x_{n}=\left\{x_{1},...,x_{n}\right\}$ into locally stationary segments by modeling each as an autoregressive process (AR). The two time intervals can be fit to an AR model of order M with coefficients $a_{m}^{i}\left(m=1,..,M\right)$ as follows:(7)$$x_{t}=\sum_{m=1}^{M}\left(a_{m}^{i}x_{t-m}+e_{t}^{i}\right)$$

The first time interval (i=1) is characterized by t∈[1,k], while the second (i = 2) by t∈[k+1,n]. This model divides the time series $x_{t}$ into two contributions: a deterministic component $x_{t-m}$ and a non-deterministic one $e_{t}^{i}$. The background noise is represented by the non-deterministic component and can be posed as a Gaussian model whose mean and variance result to be, respectively, $E\{e_{i}^{n}\}=0$ and $E\left\{{\left(e_{i}^{n}\right)}^{2}\right\}=\sigma_{i}^{2}$. In agreement with [44], the model with the lowest value of the Equation (Equation 6) results to be the one with the best division of the time series among all the competing models. The minimization of the AIC equation in (Equation 6) requires the definition of the likelihood function *L*. Considering the time series expressed in (Equation 7) with the non-deterministic part modeled as a Gaussian function, the likelihood function can be defined as in Equation (Equation 8) for the two time series i=1,2:(8)$$L\left(x;k,M,\Theta_{i}\right)=\prod_{i=1}^{2}{\left(\frac{1}{\sigma_{i}^{2}2\pi }\right)}^{\frac{n_{i}}{2}}e^{\left(-\frac{1}{\sigma_{i}^{2}}\sum_{j=p_{i}}^{n_{i}}{\left(x_{j}-\sum_{m=1}^{M}a_{m}^{i}h_{j-m}\right)}^{2}\right)}$$

In Equation (Equation 8), $\Theta_{i}=\Theta \left(a_{1}^{i},...,a_{M}^{i},\sigma_{i}^{2}\right)$ represents the parameter of the model, being that $\sigma_{i}^{2}$ is a function of the point *k*. In addition, the parameters $p_{i}$ and $n_{i}$ are functions of the extremities of the two time series that can be expressed as $p_{1}=1$, $p_{2}=k+1$, $n_{1}=k$ and $n_{2}=n-k$. In order to minimize the AIC Equation (Equation 6), the logarithm of the likelihood function (Equation 8) should be maximized. The maximum is obtained by imposing equal to zero the derivative of ln(L):(9)$$\frac{\partial \ln L(x;k,M,\Theta_{i})}{\partial \Theta_{i}}=0$$

Thus, it results in the following:(10)$$\begin{array}{c} \ln L\left(x;k,M,\Theta_{i}\right)=-\frac{n_{1}}{2}\ln\left(\sigma_{1,max}^{2}\right)-\frac{n_{2}}{2}\ln\left(\sigma_{2,max}^{2}\right)-\frac{n}{2}\left(1+\ln2\pi \right)=\\ =-\frac{k}{2}\ln\left(\sigma_{1,max}^{2}\right)-\frac{n-k}{2}\ln\left(\sigma_{2,max}^{2}\right)+C \end{array}$$
where the term *C* is a constant. Substituting Equation (Equation 10) in Equation (Equation 6), the AIC can be retrieved as function of the parameter *k*:(11)$$AIC\left(k\right)=\ln\left(\sigma_{1,max}^{2}\right)+\left(n-k\right)ln\left(\sigma_{2,max}^{2}\right)+2C$$

The equation represents a measurement of the model fitting of the two time series. Therefore, the value of *k* for which the Equation (Equation 11) is minimized represents the best fitting of the model on the two data series, as reported in Figure 2. Consequently, it represents the most probable point at which the two datasets are divided between the noise and the signal. The point found is nothing more than the onset time. One of the greatest advantages of the method presented is the possibility of calculating the accuracy of the AE damage location. Knowing the relationship AIC(k) (Equation 11), it is possible to calculate the certainty parameter DD [45] as the second derivative of the AIC function in correspondence to the onset time:(12)$$DD=\frac{AIC\left(k_{min}-\delta k\right)+AIC\left(k_{min}+\delta k\right)-2AIC\left(k_{min}\right)}{{\left(\delta k\right)}^{2}}$$
in which δk represents a small increment of time with respect to the point in which the AIC equation is minimized, $k_{min}$. In general, a bigger value of the DD parameter implies a higher precision in the determination of the onset time. The possibility of calculating the DD allows a check to be made on the detected time validity. This represents a great advantage with respect to the majority of automatic onset time detection for which the accuracy can be calibrated only before starting the test without the opportunity to check the obtained results.

In [26], tests were conducted to establish the discriminant value of the certainty parameter DD. It was observed that the measurements can be considered accurate for values of DD>0.1. The frequency of acoustic emission signals is around 20–500 kHz. The same frequency characterizes the background noise that generally is recorded together with the actual signal. For this reason, it is not practicable in this application to eliminate the presence of background noise without significantly compromising the recorded signal. In general, signal-filtering techniques can be adopted to reduce the background noise as much as possible but without completely suppressing it. In [26], the low-pass filter [46] was proven to significantly improve the accuracy in identifying onset time for signals characterized by a DD<0.1. In contrast, sometimes the application of this technique to signals with a high DD is found to significantly reduce the accuracy of the results obtained. By defining $DD_{Filter}$ and $DD_{No-filter}$, the certainty parameter DD related to filtered and unfiltered signals, respectively, it is convenient to apply the filter only in the case when the following condition is fulfilled:(13)$$DD_{Filter}>DD_{No-filter}\;\;\;\&\;\;\;DD_{No-filter}<0.1$$

The reliability of the onset time measurements obtained by the improved AIC picker method can also be assessed by accounting for the apparent velocity of the elastic wave. Known the distance $d_{ij}$ between two sensors *i* and *j*, the apparent velocity of the elastic wave $v_{ij}$ can be calculated from the onset times $t_{i}$ and $t_{j}$ related to the same event:(14)$$v_{ij}=\frac{d_{ij}}{|t_{i}-t_{j}|}$$

By knowing the propagation velocity of an elastic wave in the analyzed medium, it is possible to distinguish false positives. Indeed, by obtaining different values than the theoretical velocity, it can be inferred that the onset times obtained turn out to be false. Finally, the certainty parameter DD is used together with the apparent velocity $v_{ij}$ to determine which sensors are affected by inaccurate measurements. Such sensors are eliminated from the system of Equation (Equation 4) to increase the accuracy of fracture localization. The $v_{ij}$ is calculated for each sensor couple and DD for each measurement. The combination of these two parameters is used to assign a reliability score for the onset time determined by each sensor. Based on the calculated scores, the onset times for the sensors with the worst rating are excluded from the calculation of the fracture location. This method is applicable only if the test is carried out with redundancy of sensors, meaning more than 5.

### 3.2. CRNN for SED

The purpose of automatic SED methods is to identify the events occurring in an audio signal and the onset and offset of the sound event [47,48]. Basically, the goal is to recognize in what temporal instances different sounds are active within an audio signal. The most popular method for handling the sound event detection problem is supervised learning [49], in which an acoustic model is learned using a training set of audio recordings and their reference annotations of classroom events. The annotations provide a binary description of the temporal activity of each target sound class, indicating whether or not a class is active for each time unit. In this paper, the most popular SED techniques are applied to AE signal time series. The main idea is to exploit the similarities between these two application fields and the capabilities of the tools developed for SED to analyze acoustic emission signal data. The creation of automated systems for the detection of sound events is hindered by many challenges, some related to the characteristics of the sounds to be detected and how they appear in real environments, and others related to the actual data collection and annotation procedures. All these define the difficulties that machine learning strategies should overcome throughout the learning process. One of the most important challenges in the field of SED is to distinguish different sound events, even in case they might be overlapped. In real life, sound events do not always occur in isolation but tend to significantly overlap with one another. The demonstrated ability of SED methods to overcome this issue can be very useful in the field of acoustic emissions. In fact, in these applications, the background noise is characterized by the same frequencies as the acoustic signal. For this reason, the application of filters to eliminate such noise is not always feasible. At this juncture, the ability to distinguish the elastic wave signal from background noise can represent a considerable advantage. In this work, a convolutional recurrent neural network (CRNN) is applied for the identification of acoustic emission signals [50]. This type of neural architecture allows exploiting the combined capabilities of convolutional neural network (CNN), recurrent neural network (RNN), and fully connected layer (FC). This neural architecture consists of convolutional and recurrent layers that provide a specific role in the identification and classification of acoustic signals. The convolutional neural layers operate the feature extraction, with the goal of learning the discriminating features through consecutive convolution operations and nonlinear transformations applied to the time-frequency representation of the signal. At the same time, the recurrent layers are necessary to learn the temporal dependencies in the sequence of features presented in their input by the convolutional layers.

## 4. SED for Acoustic Emissions Onset Time Detection

In the current study, an automatic detection method was analyzed based on a convolutional recurrent neural network, coming from the context of the sound event detection (SED) [47,48]. In the literature, different neural architectures have been proposed for the SED, such as feed-forward neural networks [51,52] or LSTM [9]. In this work, a CRNN for SED is adopted, being one of the most extensively used methods for sound event detection nowadays [47,48,50].

In the following sections, the adopted SED neural network and the necessary dataset preparation are described in detail.

### 4.1. Implemented SED Model Architecture

SED is the task of recognizing the sound events and their respective temporal starting and ending times in a recording. A widely used network architecture for sound event detection is the convolutional recurrent neural network (CRNN), suitable for tasks in which temporal sequence modeling is advantageous. CRNN combines the potentials of a convolutional neural network (CNN), a recurrent neural network (RNN), and a fully connected layer (FC) in a single architecture. The convolutional layers act as feature extractors, whereas the recurrent layers aim to learn the temporal evolution of the signal. Finally, the feedforward layers have the role of producing sound event activity probabilities based on the output from the last recurrent layer.

The implemented SED model is shown in Figure 3 [47,48]. In this study, the network’s input signals are similar to mono-channel audio tracks. The input data are priorly preprocessed, as detailed in the following subsections, to provide the network with a sequence of frame-wise features. The inputs undergo three bidimensional CNN layers with 128 filters 3 × 3 rectified linear units activation functions (ReLU), followed by 1 × 5, in the first CNN layer, and 1 × 2 max pooling layers in the last two CNN layers [48]. Thereafter, the CNN outputs feed two bi-directional long short-term memory (LSTM) layers with 32 gated recurrent units (GRU) and hyperbolic tangent (tanh) activation functions [53]. Finally, two fully connected time-distributed layers are present, the foremost with 32 units, whereas the latter has a number of units as the number of output classes of the SED classification problem. In this case, to detect the onset time, the two classes are the one referred to the background noise of the signal, denoted as 0 with label ‘*No crack*’ whereas, immediately after the onset time, the AE is classified as 1 with label ‘*Crack*’. A dropout rate of 0.5 is set after each layer. The SED model was implemented in Python Keras [54] and a detailed summary is reported in Table 1.

It is worth recalling that in the last layer, the network provides through a softmax the probabilities of each time frame to belong to the various output classes, i.e., 0—no crack or 1—crack. In the SED implementation of [48], the probability threshold was set equal to 50%, meaning that the output predicted class of each time frame is the one whose probability overcomes 50%. However, since in the current problem, class 0 actually represents the background noise, whereas class 1 represents the acoustic emission, and because the goal of the problem is to properly identify the onset time instant, the threshold can be seen only for the output probabilities of class 1—crack. Therefore, every sample whose probability is lower than the user-defined threshold will be classified as belonging to class 0—no crack. In the following, the authors identified the optimal probability threshold regarding the values which optimize the error-considered metrics, i.e., the mean absolute error (MAE) and the root mean square error (RMSE) between the predicted and the true onset time instant:(15)$$\mathrm{MAE}=\frac{\sum_{1}^{n}\left|t_{true,i}-{\hat{t}}_{predicted,i}\right|}{n}$$
(16)$$\mathrm{RMSE}=\sqrt{\frac{\sum_{1}^{n}{\left(t_{true,i}-{\hat{t}}_{predicted,i}\right)}^{2}}{n}}$$
in which *n* is the total number of signals considered, $t_{true,i}$ represents the true onset time, and $t_{predicted,i}$ the onset time predicted with the machine learning model. The MAE and RMSE metrics are both expressed in seconds, and even if their informative content is apparently similar, the RMSE delivers also information about the dispersion of the error distribution since the square operation before the square root operation, thus indicating the variance of the frequency distribution of the errors. Therefore, MAE will be always less than the corresponding RMSE, and the ideal situation is when the two error metrics are minimum and with the same values. In order to compare RMSE for different variables, items, or groups, and to simplify its interpretation, the scientific literature [55,56,57,58] suggests using also the normalized RMSE (NRMSE). Despite it presenting various definitions, the most common normalization factor is the mean of the true observations [55,58], denoted as ${\bar{t}}_{true}$. Thus, the NRMSE delivers a dimensionless RMSE-related parameter defined on a common scale which permits a direct comparison and interpretation:(17)$$\mathrm{NRMSE}=\frac{\mathrm{RMSE}}{{\bar{t}}_{true}}$$

### 4.2. Datasets Description

In this study, the network’s input signals are similar to mono-channel audio tracks. As illustrated in Figure 4, the network receives inputs in terms of spectrograms in order to consider intrinsic features both in the time and the frequency domain. The vertical axis of spectrograms represents the frequency components contained in the signal, whereas the horizontal axis represents the time dimension. Finally, the information about the frequency component intensity is represented through a color-map representation: starting from the dark blue color, it represents the lowest intensity frequency components, whereas the red color indicates the highest intensities. To perform the training for the SED-based network approach, the spectrograms were calculated on the time series based on the short Fourier Transform method [59], i.e., a discrete Fourier transforms (DFT) over short overlapping windows of length equal to 32 samples and overlapping every 16 samples (denoted as hop_length), and finally converted to a dB-scaled spectrogram [60]. Therefore the time duration in terms of number of samples of the original signals is converted into time frames in the spectrograms (the single frame is denoted as *t* in Figure 4). The number of resulting frames $N_{f}$ can be calculated starting from the total number of samples in the original signal $n_{s}$ as $N_{f}=n_{s}/hop\_length$. An example of a spectrogram of a seismic signal is depicted in Figure 4. In the view of supervised learning, the input data are labeled in the following way: the ‘*no crack*’ label is assigned from 0 until the onset time, while the ‘*crack*’ label is assigned from the onset time until the duration of the signal is covered.

In the current study, to train the SED network model for AE onset time detection, seismic signals were adopted because of the parallelism with AE signals. Indeed, an earthquake can be interpreted as an example of AE on the large-scale Earth’s crust crack. The seismic signals dataset was collected from the ITACA database, i.e., the Italian Accelerometric Archive. Specifically, 410 signals were considered sampled at 200 Hz, referred to seismic station recordings coming from Northern Italy. The seismic signals resemble AE only in form however, as these signals revealed different orders of magnitudes with respect to real AE. Therefore, the signals were normalized in amplitude and simply rescaled in time, trivially considering a new sampling frequency in accordance with AE, i.e., equal to 1000 kHz [35]. The signals in the resulting database had no indication whatsoever of the onset time of the signal. For this reason, labeling was carried out manually by the users. For a massive dataset, it would be virtually possible to efficiently automatize the labeling procedure, leveraging the method of [26] presented in Section 3.1. As illustrated in Figure 4, the input signals are combined in a single track both for the training and the test set. In reality, their spectrograms (i.e., their features) were combined in a single input track. The SED network will split the input long sequence of spectrograms according to the first CNN layer input dimension equal to 256 time frames.

In order to evaluate the effectiveness of the SED neural network method on a real acoustic emission dataset, three different SED models were tested on real-world AE data publicly available in [35]. In that work, the authors provided AE data based on pencil lead break (PLB), i.e., an alternative way to artificially create AE signals, useful for onset time detection and damage localization purposes. The test was conducted by breaking a 0.3 mm HB pencil lead on the surface of a simply supported concrete beam. The PLB test was performed three times, delivering three measurement sessions denoted as A, B, and C according to the various positions of the pencil break test. Each measurement session was monitored with 10 sensors, whose locations are reported in [35]. The AE signals were acquired in Volt units through piezoelectric sensors located on a simply supported concrete beam with a sampling rate of 1000 kHz. The AE signals have a total duration of 1024 time samples, and the onset time event has a constant pre-triggering of 256 μs.

It is worth noticing that, in this preliminary study, the splitting of the dataset between training, validation, and test sets was performed on the basis of the authors’ experience. Furthermore, a manual search for a reasonable combination of the model’s hyperparameters was used since it represents a valid and extensively adopted approach [61]. Several strategies for improving neural model performance through the optimal choice of dataset partitioning and ANN hyperparameters are available in the literature. In general, the cross-validation approach is used to partition the dataset [62,63] into training, validation, and test sets. The grid-search approach [64] is one of the most extensively utilized techniques for optimizing hyperparameter selection. The application of these methodologies is beyond the scope of this study, whose primary goal is to make use of the similarities between acoustic emissions, seismic signals, and sound signals to improve the performance of the machine learning techniques applied to the acoustic emission field.

## 5. Results and Discussion

### 5.1. SED Model Trained on Seismic Data Only

The first SED neural network model studied was trained using only the seismic signals obtained as mentioned in Section 4.2 as the training and validation sets. In this scenario, the purpose was to determine if training on a different domain, i.e., the seismic dataset, provides adequate and enough accurate assessment of crack onset time in AE signals. The method exploits the similarity between the two signal types, as well as the robustness of the utilized neural architecture in classifying acoustic signals in domains other than the training domain. The seismic dataset containing 410 signals was further divided into 328 (∼80% of the dataset) in the training set and 82 (∼20% of the dataset) in the test set. The SED model was trained on seismic signals for 60 epochs, as demonstrated in the convergence curves in Figure 5, which evidence that no overfitting issues arose since the validation loss evidenced a global descending trend after epoch 10.

In a first stage, the SED model trained on seismic signals was tested directly on the test set of 82 seismic signals to verify the effectiveness of the training process. The main results are reported in Figure 6. The left graph illustrates the trends of the MAE and RMSE in terms of onset time predictions with respect to the probability threshold levels. The optimal threshold level was set in correspondence of the the minimum value of the error curves, in this case it corresponds to the 96%. The right graph depicts the onset time prediction for all of the 82 seismic signals of the test set for the optimal found probability threshold level. When the blue curve assumes value zero, the first time frame of that test signal is classified as 1—crack, thus indicating that the onset time is not properly identified because it corresponds to the first time frame. From this graph, the seismic data test set revealed a consistent good agreement between the predictions. Since the SED is a classification problem, the quantitative results to assess the test set predictions’ quality are reported Table 2, illustrating the confusion matrix and other evaluation metrics in terms of precision, recall and f1-score, which is the harmonic mean of precision and recall:(18)$$f1-\mathrm{score}=\frac{1}{\frac{1}{\mathrm{precision}}+\frac{1}{\mathrm{recall}}}$$

The evaluation metrics approach 98%, demonstrating the successful training process along the 60 training epochs, whereas the overall classification accuracy of the trained SED model on the seismic data test set is equal to 97.685%. In Figure 5, the receiver operating characteristic (ROC) curve is reported, which illustrates that the area under the ROC curve (AUC) value is really close to the ideal unitary value [65].

In a further stage, the SED model trained on the seismic signal was tested on AE signals provided in [35], as described in Section 4.2. The MAE and RMSE error curves along the probability threshold are reported in Figure 7, in the upper pane. The optimal threshold levels were identified as 85%, 90%, and 96%, respectively, for the three PLB test data provided in [35] denoted as A, B, and C. Each test contains 100 signals in total. However, the comparison between the onset time predictions and the true values, Figure 7, in the lower pane, revealed poor performance of the SED trained on seismic data only and tested on AE data. The same conclusions can be retrieved focusing on Table 3. The confusion matrices highlighted a general biased behavior of the SED model to classify most of the time frame as 1—crack, without permitting any correct evaluation of the onset time, corresponding to the pre-triggering time of 256 μs. It is worth mentioning that in this preliminary investigation, the neural network was just trained using seismic signals before being tested with acoustic emission signals. As a result, the outcomes of this approach are influenced by the clear difference between the training and test sets. The method can be enhanced in the future by employing domain adaption techniques [66,67], which allow for better generalization of the neural model and hence increase its efficiency when used in a domain other than the one on which it was trained.

### 5.2. Training SED Only on AE Dataset

The second SED model was trained and evaluated directly on the dataset of AE signals provided by [35] as described in Section 4.2. The dataset, which was obtained by a point lead break test on a concrete cube, consists of 300 acoustic emission signals subdivided into three sub-datasets A, B, and C, each of which contains 100 individual signals. In this scenario, the network training was performed with only the 100 signals from dataset A and then tested on datasets B and C. The goal of this model was to demonstrate the usefulness and robustness of the chosen SED network even with a restricted dataset, i.e., limited in number of available data. Similarly to the previous section, the current SED model was trained on AE of PLB test A signals for 60 epochs in total, as demonstrated in Figure 8. The training curves evidenced that no overfitting issues arose since the validation loss evidenced a global descending. The trained model was then tested on PLB test set B and C as depicted in Figure 9. In the upper pane, the error curves of the MAE and RMSE on the onset time prediction are reported in the function of the probability threshold. With the test B, the optimal threshold was detected at 99%, whereas for the PLB test C, it was defined as 98%. In the lower pane of the figure, the onset time predictions are compared with the true constant onset time equal to the pre-triggering time of 256 μs. It is worth recalling that when the blue curve stalls on zero as the predicted onset time, this means that the first time frame is classified as 1—crack. In general, a fairly good behavior was evidenced from these graphs, with a worse performance of the test C than the test B. The quantitative evaluation of the SED onset time predictions are reported in Table 4. The confusion matrix supported the qualitative behavior highlighted in Figure 9, demonstrating that the overall accuracy of the PLB test B was 95.965%, better than the PLB test C, which reaches 94.280%. In Figure 8, the ROC curves are reported both for PLB test set B and C, demonstrating even in this case that the AUC values are close to the ideal unitary value in both cases [65].

### 5.3. Fine-Tuning SED on AE Dataset

For the purpose of improving the performance of the SED trained on seismic signals described in Section 4.2, to properly detect the onset time of real data AE, a final fine-tuning approach is proposed. The authors’ intention, in this case, was to demonstrate the possibility of exploiting the similarity between seismic signals and acoustic emissions to increase the training dataset. Indeed, it is expected to demonstrate an increase in model efficiency due to the ability to run training with a larger dataset, even if it belongs to a different class of signals than that of the application. At the same time, it is intended to demonstrate how the precision of the neural model compared to the first scenario is significantly increased by fine-tuning using a limited dataset of acoustic emission signals. In fact, the first phase of training allows the network to perform a pre-training in order to begin distinguishing the parts of the series temporal where a signal is present from those where one is not. In contrast, the second phase allows the network to specialize in the task of identifying certain signals, such as audible emissions. This stage improves the model’s efficiency by taking into account the peculiarities of the signals on which the final model will be used, such as the signal-to-noise ratio, the frequency of the signal, the shape of the signal over time, and other factors.

In this section, the authors adopted the sub-optimal weights of the model trained on seismic signals only, and training the model for further 60 epochs using as training set the AE data coming from the PLB test A [35]. The training convergence curves are reported in Figure 10. These curves appear smoother that the convergence curves of SED model in Section 4.2, and still highlight the absence of overfitting issues. The trained model was then tested on PLB test sets B and C as depicted in Figure 11. In the upper pane, the error curves of the MAE and RMSE on the onset time prediction are reported in the function of the probability threshold. With the test B, the optimal threshold was detected at 95%, whereas for the PLB test C, it was defined as 97%. In the lower pane of the figure, the onset time predictions are compared with the true constant onset time equal to the pre-triggering time of 256 μs. It is worth recalling that when the blue curve stalls on zero as the predicted onset time, this means that the first time frame is classified as 1—crack. In general, a fairly good behavior was evidenced from these graphs, with a better performance of the test B than the test C. The quantitative evaluation of the SED onset time predictions are reported in Table 5. The confusion matrix supported the qualitative behavior highlighted in Figure 9, demonstrating that the overall accuracy of the PLB test B was 96.370%, better than the PLB test C, which reaches 92.383%. The fine-tuned SED demonstrated the ability of the model to improve the classification performance with respect to the model trained on seismic data only. Moreover, it is worthwhile observing that for the test B, the fine-tuned SED performed better than the SED trained on AE only, which reached a less overall accuracy of 95.965%. On the other hand, focusing on PLB test C, the fine-tuned SED delivered a less global accuracy than the SED trained on AE only in the previous section, which reached a greater overall accuracy of 94.280%. Finally, in Figure 10, the ROC curves are reported both for PLB test set B and C for the fine-tuned model, illustrating even in this case that the AUC values are close to the ideal unitary value in both cases [65].

### 5.4. Discussion

Three neural models were examined in this chapter for determining the onset time of AE signals. All of the models examined were built on an architecture geared for sound event detection. The main difference between the models is related to the different datasets on which they were trained. In the first scenario, only seismic signals were used as the training set. The goal of this study was to show the robustness of the models utilized in functioning in data domains other than the training domains. The model proved to be perfectly capable of identifying onset time related to seismic signals as demonstrated in Table 2. Such performance was even improved by going so far as to consider classifying in the “1—crack” category only those items that had a probability above a certain threshold of belonging to that category. In other words, it was determined to create an asymmetric classification in which all the points with a probability less than the stated threshold were classified as “no crack”. This novel strategy improved the model’s performance, as illustrated in Figure 6. This novel technique was also beneficial for testing the network on AE signals. In this scenario, the difference in signal-to-noise ratio between seismic signals and AEs hampered the model’s ability to correctly categorize AE signals. In reality, in the test on AE signals, the neural network was unable to distinguish between signal and background noise points, and almost all of the points studied were labeled as “1—crack” as shown in Table 3. The second neural network was trained using a dataset of 100 AE events. The goal in this scenario was to demonstrate the SED architecture’s potential to achieve outstanding outcomes despite training on a limited dataset. As demonstrated in Table 4, the model performed admirably in this scenario. Again, the threshold strategy for asymmetric signal categorization improved model performance significantly by allowing for improved MAE and RMSE evaluation metrics. In fact, these metrics are more than halved with the application of the new method, as shown in Figure 9. In the last case tested, a fine-tuning of the model previously trained on the dataset of only seismic signals was performed. Fine-tuning was carried out by performing 60 more epochs of model training on a dataset made up of only 100 AE signals. In this case, the goal was to test whether it is possible to improve the performance of the SED model in the case of having a limited training dataset. Indeed, seismic signals are utilized to pre-train the model, allowing it to learn to recognize signals early on and subsequently specialize on the type of signal under consideration, in this case, AE signals. In terms of precision, recall, and F1-score, the suggested model performs quite similarly to the model trained solely on AE data, as shown in Table 5. Even though the two models appear to behave similarly, the improvement in pre-training efficiency using the dataset of seismic signals can be seen in the MAE and RMSE metrics. These metrics are the most essential valuation metrics for the problem under consideration. Indeed, the proposed method’s goal should be to determine the onset time with the greatest precision possible. Precision in determining the correct onset time is essential for accurate crack localization. As a result, obtaining low MAE and RMSE values is more significant than obtaining high values for traditional neural model evaluation metrics. Furthermore, given the high speeds at which the AEs propagate inside the concrete medium (about 4000 m/s), even a minor improvement in determining the onset time results in a significant improvement in crack localization. In this case, using pre-training on a collection of seismic signals resulted in overall improvements in precision in determining the onset time. Finally, it is evident how the third model outperforms the first two, even because the optimal threshold value is lower. This is an indication of the network’s superior ability to classify the test set. The threshold values used in this study have been optimized over the MAE and RMSE. As previously stated, the goal is to find a model that minimizes these quantities for a more precise determination of the onset time. In general, it is seen that models perform better with threshold values greater than 95%. It is also true that very high threshold values may produce overcautious results when used with models trained on larger scale datasets. According to the authors, in the case of using the model’s field, the optimal value on which to set the threshold rises to around 85–90%. Finally, focusing on the NRMSE metrics, it is possible to compare all the trained models based on a common unitary error scale. It is worth noting that the model trained on seismic data produced the best results in terms of NRMSE since there was no domain knowledge transfer or any domain adaptation. As a matter of fact, the test set was always of the same nature as the training seismic signals, thus providing an NRMSE of 14.21% as reported in Table 2. On the other hand, the model trained only on AE data provided a slight increase in the normalized error, as illustrated in Table 4 with an NRMSE of 21.88% for test B and 34.73% for test C. However, when the domain knowledge was transferred with the fine-tuning approach, the onset time identification for AE data provided a slight decrease of the normalized error, as evidenced in Table 5 with an NRMSE of 19.60% and 34.26% for test B and C respectively.

## 6. Conclusions and Future Developments

In this paper, two methods for onset time identification of acoustic emission signals are presented. The first method is one of the most accurate in the literature and involves the application of the Akaike Information Criterion so as to identify two time series in the signal, one formed by the background noise alone and one formed by the signal itself. Because this strategy is focused on minimizing the AIC function, it can only be applied to time series with a single signal. As a result, data processing is required to identify and distinguish the time series associated with each recorded signal. In practical terms, the procedure begins with the use of a dynamic threshold method to determine a first-attempt onset time. The approach is then applied to the whole time series prior to the first-attempt onset time determined in this way. This gives a measure of a second-attempt onset time. At this point, the procedure is applied for the third time to a time series of length 2Δk and centered on the onset time of the second attempt. After that, the final onset time is computed. To improve the procedure’s accuracy, the calculation is repeated on signals filtered with a low-pass filter. The measurement is evaluated by calculating the certainty parameter DD. Finally, the onset time with lower DD calculated for the filtered and unfiltered signals is used for the calculation of the crack (Improved AIC; I-AIC). Because acoustic emission monitoring is frequently performed in sensor redundancy compared to the minimums required to compute fracture position, the precision of this approach can be further improved by eliminating measurements considered as less reliable. By this method, the signal is processed several times to improve the precision of the measurement. As a result, the approach appears to be more suited for data post-processing applications than real-time applications. This method originally presented by some of the authors is here reconsidered and automatized for use in a new SW for post processing data analysis of AE. In particular, due to the high accuracy of I-AIC and the necessity to perform this localization in the post-processing phase, this procedure will be used in an off-line way and for those cases in which the precision of the results has to be maximized with respect to the final relative error (percentage of events with a deviation >5 mm from the exact locations for each coordinates axis less than 3%) [26]. The second provided strategy appears to be more appropriate for the purpose of the real-time application. This method is based on the use of a convolutional recurrent neural network designed for sound event detection. This neural architecture was used to compare three independent models, each of which was trained on a different dataset. The resemblance between seismic signals, acoustic emission signals, and sound waves was used in the first scenario. Indeed, a model for sound wave categorization was trained on a dataset of seismic signals. This model was then evaluated on an acoustic emission signal dataset. In this case, the goal was to test the robustness of the proposed method. In fact, the effectiveness of a model trained on different signals than those of the application was investigated. In the second scenario, however, the model was trained directly on a small sample of acoustic events. In this scenario, the goal was to put the method to the test in order to determine its effectiveness even in the usual case of limited datasets. Finally, in the third case, the added model was fine-tuned using only statistical signals. The fine-tuning procedure was thus carried out using a limited dataset of acoustic emission signals. The goal of this model was to see if it was possible to improve the efficiency of the CRNN SED by using a dataset other than the one used for application in the case of a limited training dataset. Once the three neural models are obtained, an innovative method based on defining a probability threshold that increases the likelihood that a specific point in a time series belongs to a signal rather than a background rumor is defined. The output of the neural network is the probability, as defined by the neural model, that a given point belongs to one category rather than another. In this case, there are two distinct categories: “0—no crack” and “1—crack’,’ where belonging to the first implies that the point is part of the background noise, whereas belonging to the second implies that the point is part of the signal. In the case of the studied problem, it is much more likely that a point will be incorrectly classified as part of the signal despite being only a component of the background rumor. To that end, in order to improve the accuracy of the proposed method, a minimum probability, bigger than the standard, of belonging to the category “1—crack” is set in order to designate a point to that category. Different threshold values were tested for the proposed models in order to determine which value produced the best results. Among the various tests, the values that would allow for lower MAE and RMSE values were identified. These two metrics are crucial for assessing the precision of the models proposed for determining the onset time of the signal. Indeed, the idea is to select the model with the lowest MAE and RMSE values rather than the one distinguished by high precision, recall, and F1-score values. Indeed, for the problem under consideration, a model that identifies the exact onset time point fewer times but achieves a lower overall error is preferable. Indeed, this error will affect the accuracy of crack localization. In general, the first model was found to be the least reliable of the three. In fact, the difference in the ratio of background noise amplitude to signal amplitude between seismic signals and acoustic emissions causes the model to fail to correctly classify the signal. Instead, the second and third models, due to the training performed on a limited number of acoustic emission signals, are found to be the best performing. In terms of precision, recall, and F1-score, these two models produce extremely similar results. Nonetheless, the third model is the one with the lowest MAE and RMSE. As can be observed, boosting the training set by using seismic signals gives an improvement in the final results. Although this improvement may appear minor in absolute terms, it can result in a large improvement in crack location accuracy. Given the high speeds at which acoustic emissions travel through the material, even little increases in onset time identification accuracy can result in considerable gains in fracture localization. The method tested has the added advantage of being scaled-up since large datasets composed of seismic signals are freely available online. Therefore, it can be considered to improve the proposed method by pre-training the model on a larger dataset. In conclusion, the CRNN [47] appears to be the more advantageous solution for all scenarios, requiring a big amount of data and a real-time localization procedure. This is because, in addition to achieving extremely close to ideal values for the evaluation criteria, the network is theoretically capable of identifying the cracking mode. Furthermore, the CRNN contained in the sound event detection framework handles the problem of determining the time of onset as a classification problem. If one were to directly ascribe the labels ‘No Crack’, ‘Mode I’, and ‘Mode II’ during data labeling, the network would provide the onset time and offset time of the event together with the crack type. The presented method is shown to be a viable alternative to traditional techniques. The proposed method is proven to provide several advantages. Actually, it is successfully demonstrated that data and techniques from other application domains can be applied to acoustic emission analysis. In particular, the broadening of the training dataset with seismic accelerograms improves the accuracy of onset time prediction. Furthermore, deep learning techniques designed for sound event detection are proven to be effective in detecting onset time. Finally, the next steps in the development of the presented work will be the implementation of such a technique for the classification of the fracture’s mode, as well as the application and testing of the method on experimental data. Laboratory tests on concrete samples will be used to obtain such data in order to classify the precision level by changing the material, the testing set, and the certainty parameters ruling the filtering of the signals and the results for the I-AIC and the thresholds governing the CRNN. In the future, an interesting research topic could be comparing different neural models designed for SED to determine the most accurate model in determining the onset time of acoustic emissions. This comparison could also take into account the accuracy loss that might occur if simpler and lighter neural models are used. Indeed, it may be interesting to define the smallest neural model that allows for a certain level of precision. This model would offer the opportunity to be installed directly in piezoelectric sensors. This would allow the definition of the onset time directly at the sensor level, reducing the amount of data to be transferred from the sensors to the central processing unit. Finally, one of the possible future developments may be to examine the robustness of the tested models in comparison to the background noise in order to determine which results are the most effective for use in real-world applications.

## Figures and Tables

**Figure 1 sensors-23-00693-f001:**
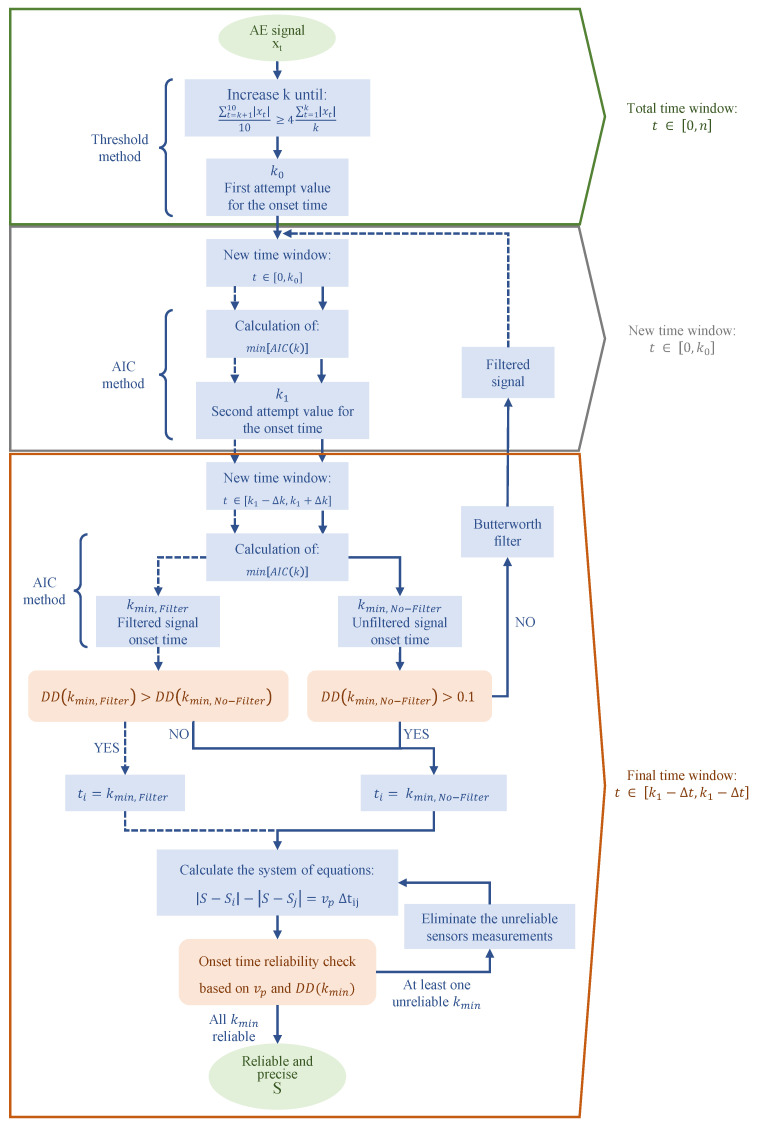
Improved AIC picker workflow.

**Figure 2 sensors-23-00693-f002:**
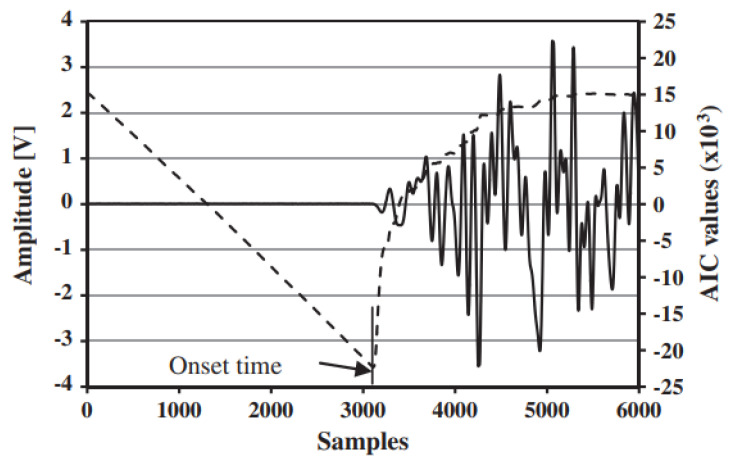
AIC function.

**Figure 3 sensors-23-00693-f003:**
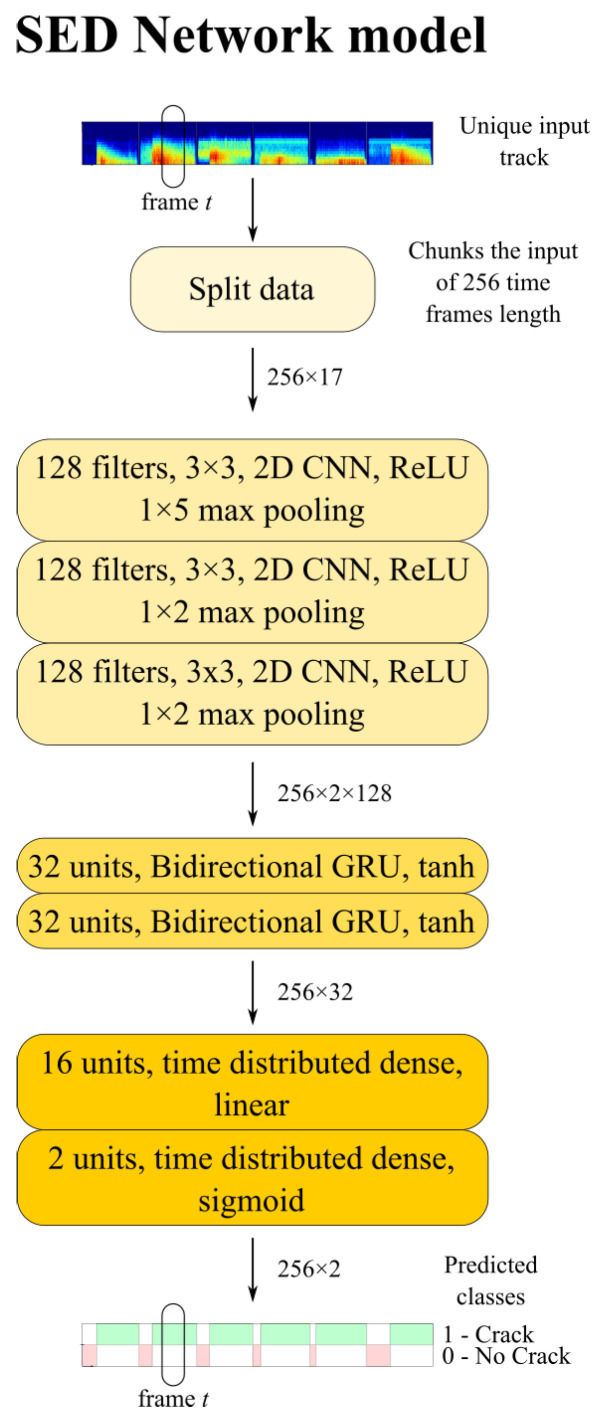
CRNN for SED.

**Figure 4 sensors-23-00693-f004:**
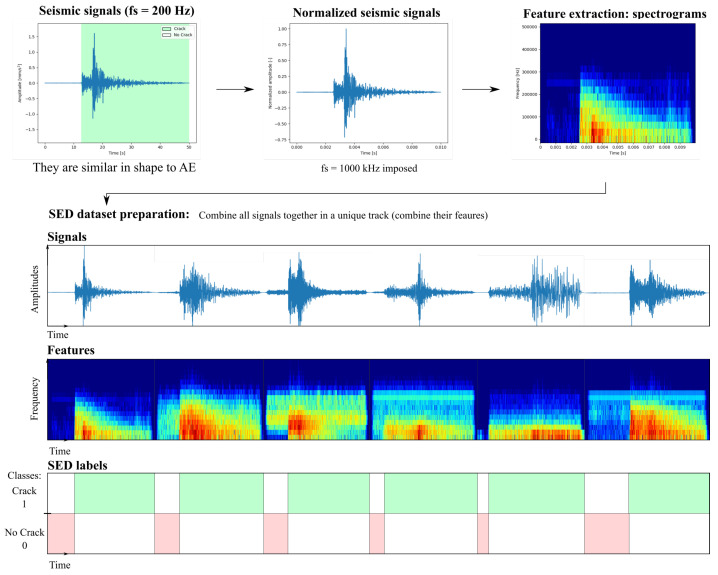
SED data and feature preparation.

**Figure 5 sensors-23-00693-f005:**
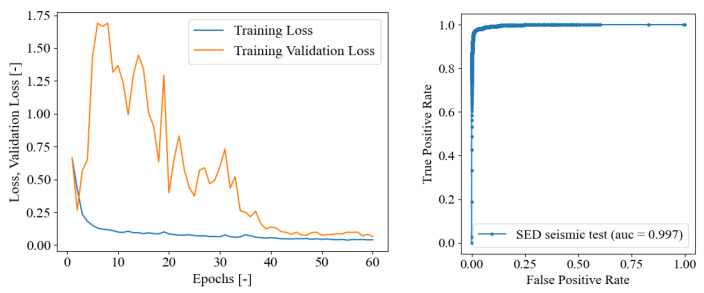
On the **left**: SED trained on seismic data only, training loss convergence curves. On the **right**: ROC curve for seismic test set.

**Figure 6 sensors-23-00693-f006:**
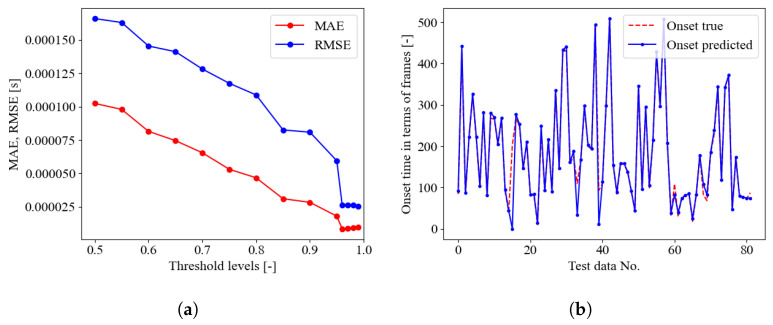
SED trained on seismic data only. (**a**) Crack class probability threshold evaluation in terms of MAE and RMSE; (**b**) onset time predictions in terms of frames on the test set (82 seismic data) for optimal threshold equal to 96%.

**Figure 7 sensors-23-00693-f007:**
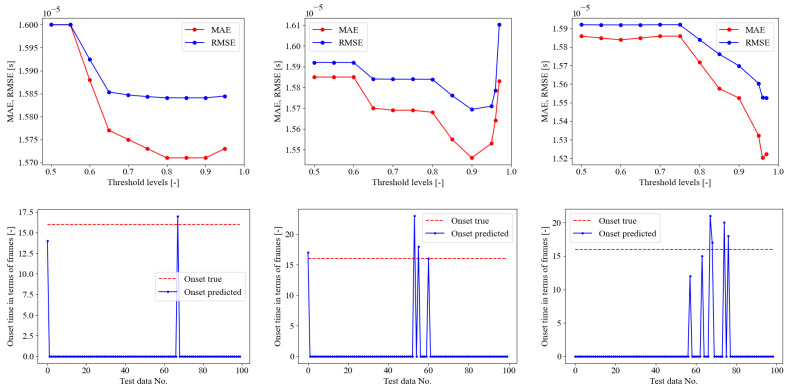
SED trained on seismic data only. Upper pane, crack class probability threshold evaluation, lower pane onset time predictions in terms of frames on the test set (82 seismic data) for optimal threshold. From left to right, PLB test A (optimal threshold of 85%), B (optimal threshold of 90%), and C (optimal threshold of 96%).

**Figure 8 sensors-23-00693-f008:**
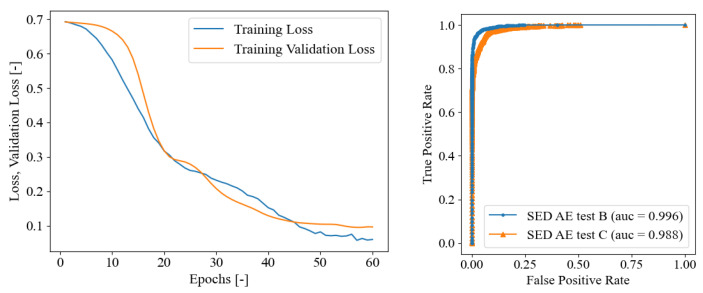
On the **left**: SED trained on AE only (training set: PLB test A), training loss convergence curves. On the **right**: ROC curve for AE PLB test set B and C.

**Figure 9 sensors-23-00693-f009:**
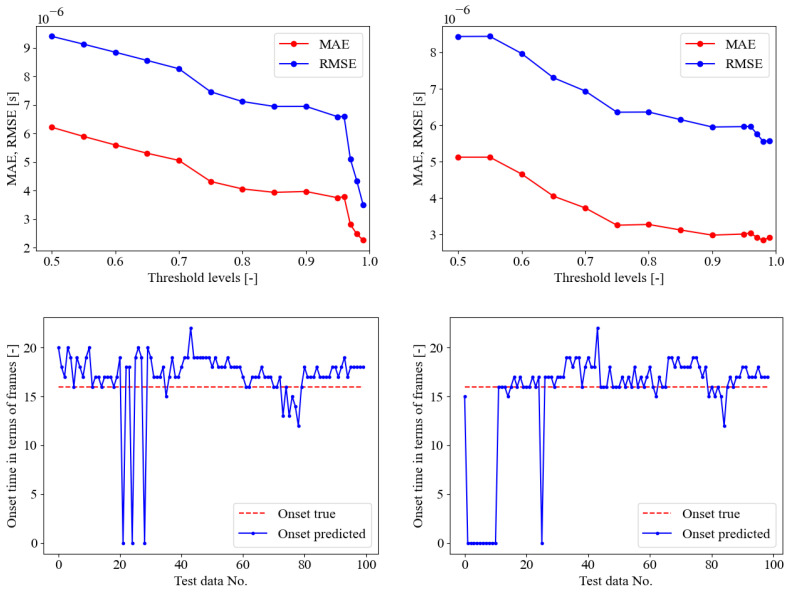
SED trained on AE only (training set: PLB test A). Upper pane, crack class probability threshold evaluation, lower pane onset time predictions in terms of frames on the PLB test sets B and C for optimal threshold. From left to right, PLB test B (optimal threshold of 99%), and C (optimal threshold of 98%). The onset time true values are always at a constant pre-trigger time of 256 μs (frame No. 16).

**Figure 10 sensors-23-00693-f010:**
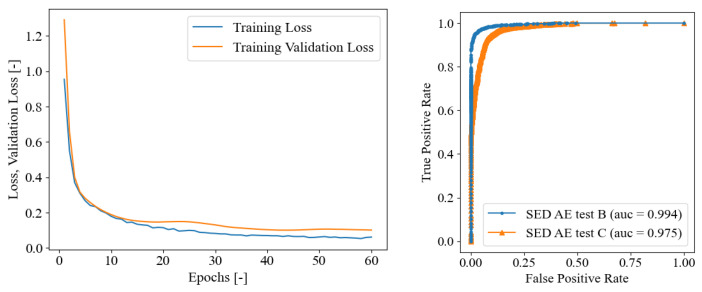
On the **left**: Fine-tuned SED, training loss convergence curves. On the **right**: ROC curve for AE PLB test set B and C.

**Figure 11 sensors-23-00693-f011:**
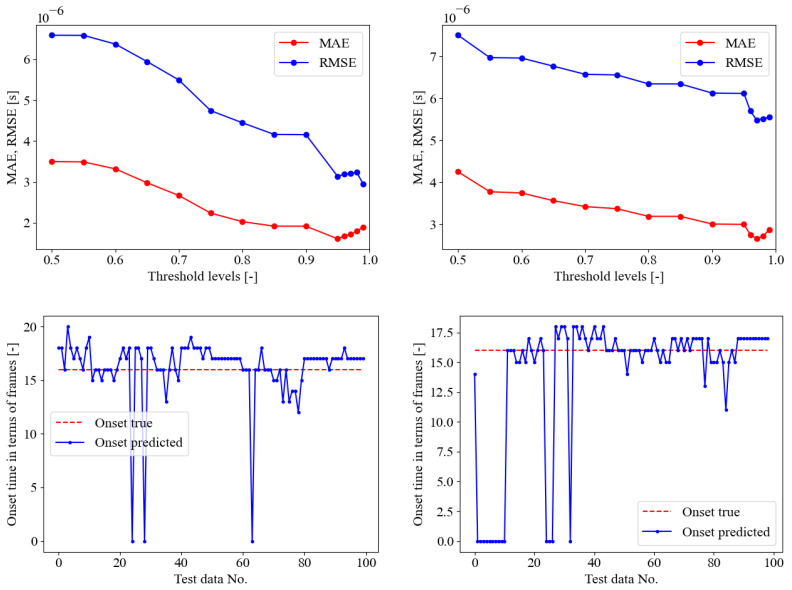
Fine-tuned SED on PLB test A. Upper pane, crack class probability threshold evaluation, lower pane onset time predictions in terms of frames on the PLB test sets B and C for optimal threshold. From left to right, PLB test B (optimal threshold of 99%), and C (optimal threshold of 98%). The onset time true values are always at a constant pre-trigger time of 256 μs (frame No. 16).

**Table 1 sensors-23-00693-t001:** SED model summary.

Layer (Type)	Output Shape	Param #
input_2 (InputLayer)	[(None, 1, 256, 17)]	0
conv2d_3 (Conv2D)	(None, 128, 256, 17)	1280
batch_normalization_3 (BatchNormalization)	(None, 128, 256, 17)	512
activation_3 (Activation)	(None, 128, 256, 17)	0
max_pooling2d_3 (MaxPooling2D)	(None, 128, 256, 4)	0
dropout_4 (Dropout)	(None, 128, 256, 4)	0
conv2d_4 (Conv2D)	(None, 128, 256, 4)	147,584
batch_normalization_4 (BatchNormalization)	(None, 128, 256, 4)	512
activation_4 (Activation)	(None, 128, 256, 4)	0
max_pooling2d_4 (MaxPooling2D)	(None, 128, 256, 2)	0
dropout_5 (Dropout)	(None, 128, 256, 2)	0
conv2d_5 (Conv2D)	(None, 128, 256, 2)	147,584
batch_normalization_5 (BatchNormalization)	(None, 128, 256, 2)	512
activation_5 (Activation)	(None, 128, 256, 2)	0
max_pooling2d_5 (MaxPooling2D)	(None, 128, 256, 1)	0
dropout_6 (Dropout)	(None, 128, 256, 1)	0
permute_1 (Permute)	(None, 256, 128, 1)	0
reshape_1 (Reshape)	(None, 256, 128)	0
bidirectional_2 (Bidirectional)	(None, 256, 32)	31,104
bidirectional_3 (Bidirectional)	(None, 256, 32)	12,672
time_distributed_2 (TimeDistributed)	(None, 256, 32)	1056
dropout_7 (Dropout)	(None, 256, 32)	0
time_distributed_3 (TimeDistributed)	(None, 256, 2)	66
strong_out (Activation)	(None, 256, 2)	0
Total params: 342,882		
Trainable params: 342,114		
Non-trainable params: 768		

**Table 2 sensors-23-00693-t002:** Confusion matrix and classification metrics for SED trained on seismic data only, tested on the seismic test set (82 earthquake signals).

	Predicted Classes		Threshold:			96%		
**True Classes**	**0 No Crack**	**1 Crack**	**Precision**	**Recall**	**F1-Score**	**MAE**	**RMSE**	**NRMSE**
0 No Crack	98.88%	1.12%	96.58%	98.88%	97.72%	[s]	[s]	[-]
1 Crack	3.51%	96.49%	98.86%	98.86%	98.86%	8.598 × 10^−6^	2.635 × 10^−5^	0.1421

**Table 3 sensors-23-00693-t003:** Confusion matrix and classification metrics for SED trained on seismic data only, tested on the AE datasets (100 signals for 3 setups A, B, and C).

**AE—Test A**	Predicted Classes		Threshold:		85%
True Classes	0 No Crack	1 Crack	Precision	Recall	F1-score
0 No Crack	4.36%	95.64%	96.77%	4.36%	8.34%
1 Crack	0.15%	99.85%	51.08%	51.08%	51.08%
**AE—Test B**	Predicted Classes		Threshold:		90%
True Classes	0 No Crack	1 Crack	Precision	Recall	F1-score
0 No Crack	8.02%	91.98%	69.31%	8.02%	14.37%
1 Crack	3.55%	96.45%	51.19%	51.19%	51.19%
**AE—Test C**	Predicted Classes		Threshold:		96%
True Classes	0 No Crack	1 Crack	Precision	Recall	F1-score
0 No Crack	9.79%	90.21%	71.81%	9.79%	17.22%
1 Crack	3.84%	96.16%	51.59%	51.59%	51.59%

**Table 4 sensors-23-00693-t004:** Confusion matrix and classification metrics for SED trained on AE only (training set: PLB test A), tested on the AE datasets (100 signals for 3 setups B, and C).

**AE—Test B**	Predicted Classes		Threshold:			99%		
True Classes	0 No Crack	1 Crack	Precision	Recall	F1-Score	MAE	RMSE	NRMSE
0 No Crack	98.93%	1.07%	93.39%	98.93%	96.08%	[s]	[s]	[-]
1 Crack	7.00%	93.00%	98.86%	98.86%	98.86%	2.260 × 10^−6^	3.501 × 10^−6^	0.2188
**AE—Test C**	Predicted Classes		Threshold:			98%		
True Classes	0 No Crack	1 Crack	Precision	Recall	F1-score	MAE	RMSE	NRMSE
0 No Crack	93.94%	6.06%	94.59%	93.94%	94.26%	[s]	[s]	[-]
1 Crack	5.38%	94.62%	93.98%	93.98%	93.98%	2.838 × 10^−6^	5.557 × 10^−6^	0.3473

**Table 5 sensors-23-00693-t005:** Confusion matrix and classification metrics for fine-tuned SED, tested on the AE datasets (100 signals for 3 setups A, B, and C).

**AE—Test B**	Predicted Classes		Threshold:			95%		
True Classes	0 No Crack	1 Crack	Precision	Recall	F1-score	MAE	RMSE	NRMSE
0 No Crack	98.30%	1.70%	94.64%	98.30%	96.43%	[s]	[s]	[-]
1 Crack	5.56%	94.44%	98.23%	98.23%	98.23%	1.610 × 10^−6^	3.135 × 10^−6^	0.1960
**AE—Test C**	Predicted Classes		Threshold:			97%		
True Classes	0 No Crack	1 Crack	Precision	Recall	F1-score	MAE	RMSE	NRMSE
0 No Crack	91.29%	8.71%	93.33%	91.29%	92.30%	[s]	[s]	[-]
1 Crack	6.52%	93.48%	91.47%	91.47%	91.47%	2.657 × 10^−6^	5.482 × 10^−6^	0.3426

## Data Availability

The data used to support the findings of this study are available from the corresponding author upon reasonable request.

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
