# Peer review of "Acoustic Emission and Artificial Intelligence Procedure for Crack Source Localization"

_sensors, 2023, doi:10.3390/s23020693_

Round 1
Reviewer 1 Report
The authors presented damage localization using acoustic emission sensor signals and machine learning techniques. The reviewer has the following concerns about the manuscript:
1. The authors should modify the article title to mention specifically crack or damage source localization. Source localization creates confusion on if the source is the sensor source or the crack/damage source.
2. The first method, the triangulation method, is used on the assumption that " elastic wave travels directly from the crack source to the acoustic sensors" and "the wave path is represented by the straight line connecting the point where the crack occurs and the sensors". Is it practical, justified and scientific to assume that for a journal paper? Please provide a robust explanation for this assumption.
3. For the second method, the authors used CRNN. Is there any specific reason why this method is used? LSTM is proven to be a better method than a combination of CNN and any vanilla RNN for sequential time-series data, as presented in this article.
4. The introduction section lacks the variability in the literature review presented in this article. It is advised that authors cite the following papers to improve the contrast and include recent crack/damage localization using AI methods.
(a) Parisi, F., Mangini, A. M., Fanti, M. P., & Adam, J. M. (2022). Automated location of steel truss bridge damage using machine learning and raw strain sensor data. Autom. Constr., 138, 104249.
(b) Sony, S., Gamage, S., Sadhu, A., & Samarabandu, J. (2022). Vibration-based multiclass damage detection and localization using long short-term memory networks. Structures, 35, 436–451.
(c) Entezami, A., Sarmadi, H., & Mariani, S. (2020). An Unsupervised Learning Approach for Early Damage Detection by Time Series Analysis and Deep Neural Network to Deal with Output-Only (Big) Data. Eng. Proc., 2(1), 17.
(d) Flah, M., Ragab, M., Lazhari, M., & Nehdi, M. L. (2022). Localization and classification of structural damage using deep learning single-channel signal-based measurement. Autom. Constr., 139, 104271.
Reviewer 2 Report
The paper is a really good one, for which I have few words to add. The authors did a really good job in designing the structure of the paper, the experiments and also in developing the idea. My only suggestions is to check the English, as it appears evident that authors are not English native speakers.
Reviewer 3 Report
The manuscript investigates deep learning method for source localization.
The work provides an in-depth introduction. Overall, I think the paper needs to undergo major revisions before publication could be considered. My comments, questions and suggestions are listed below.
Introduction:
- Authors discuss about source localization using acoustic emission signals, please cite the recent literature about standard threshold and cross-correlation techniques [10.1109/JSEN.2018.2890568, doi.org/10.3390/s20185042].
Methodology:
-Did authors used cross-validation in training models? Cross-validation is used for preventing overfitting, authors should argue which techniques have been applied in their pipeline to handle this problem. The dataset split in training/test is neither sufficient nor cautelative and could represent a lack in content and methodology.
-To compare two items or groups is suggest using also the normalized RMSE (NRMSE), broughting on the same scale different groups.
Results:
-In results, authors should help readers to better show results with the receiver operating characteristic curve (ROC) and area under the ROC curve (AUC) for each comparison.
Reviewer 4 Report
This manuscript proposed a novel acoustic emission-based sound event detection using deep learning technique, where a regional convolution neural network (RCNN) was developed for the task of interest. To evaluate the performance of the proposed method, three different datasets consisting of seismic signals and acoustic emission signals were used, with satisfactory results. Overall, the topic of this study is interesting, and the manuscript was well organised and written. I suggest that it can be accepted for publication, if the authors can well address the following comments.
1. The main innovation and contribution of this manuscript should be clearly clarified and highlighted in abstract and introduction.
2. Broaden and update literature review on CNN or deep learning in practical engineering applications. E.g. Vision-based concrete crack detection using a hybrid framework considering noise effect.
3. The performance of the deep learning models is overly related to the setting of hyperparameters. How did the authors set them in this research to achieve the optimal prediction performance?
4. More information about how the training and test data were determined should be provided.
5. The proposed method has not been proved with its advantages. A comparison with other similar deep learning models is suggested.
6. How about the robustness of the proposed method against noise effect?
7. More future research should be added in conclusion part.
Round 2
Reviewer 1 Report
The authors have considered the suggestions, and the manuscript is improved.
Reviewer 3 Report
The authors have satisfactorily addressed most of my concerns.
Reviewer 4 Report
The authors well addressed the reviewer's comments. Hence, I suggest that this revised version can be accepted for publication in Sensors.